# Association between Physical Activity Levels and Body Composition among Healthy Older Japanese Adults during a Snowy Winter: A Cross-Sectional Study

**DOI:** 10.3390/ijerph17155316

**Published:** 2020-07-23

**Authors:** Tomoko Shimoda, Teppei Suzuki, Kaori Tsutsumi, Mina Samukawa, Sadako Yoshimura, Katsuhiko Ogasawara

**Affiliations:** 1Faculty of Health Sciences, Hokkaido University, Sapporo, Hokkaido 060-0812, Japan; tshimoda@hs.hokudai.ac.jp (T.S.); tsutsumi@hs.hokudai.ac.jp (K.T.); mina@hs.hokudai.ac.jp (M.S.); sadako@med.hokudai.ac.jp (S.Y.); 2Iwamizawa Campus, Hokkaido University of Education, Iwamizawa, Hokkaido, 068-8642, Japan; suzuki.teppei@i.hokkyodai.ac.jp

**Keywords:** exercise, physical strength, aging, diet, health

## Abstract

Background: Despite a long average lifespan, increased life expectancy does not guarantee higher quality of life. Methods: To contribute in understanding some determinants of healthy life expectancies in older Japanese individuals in a snowy winter region, we investigated the indicators of health. Local residents (*n* = 124) in the city of Iwamizawa volunteered for health examinations from January 2016 to March 2016. We recorded activity via daily steps for 2-week periods. In addition, we measured body composition, grip strength, and assessed nutritional status. Results: Analysis of body composition and daily activity indicated that women who walked more than 4000 steps had lower fat mass and increased muscle mass. Men with >3.0 metabolic equivalents (METs) when walking had lower body fat. Conclusion: For healthy older Japanese individuals in this snowy winter region, walking >4000 steps daily for women and exercise of >3.0 METs for men may indicate health-promoting activities.

## 1. Introduction

Although the Japanese have a long average lifespan, increased life expectancy does not necessarily indicate a higher quality of life. Longer life expectancy may increase the risk of diseases or disability before death [1]. Healthy life expectancy is defined as the length of a person’s lifespan that is not characterized by limitation in daily activities due to health problems [2]. This measure summarizes information about lifespan, health status, and health-related quality of life [3]. The difference between life expectancy and healthy life expectancy indicates the average number of years lived in poor health. Therefore, it is important to extend healthy life expectancy, rather than merely increase life expectancy in years. The proportion of older Japanese increased to more than 7% and 14% in 1970 and 1994, respectively. The rate of the total older population has continued to increase since then, reaching 26.7% in 2016 (33.92 million) [4,5]. Currently, individuals who are 75 years and older comprise 12.9% of the total population [4]. In Japan, which is a rapidly aging society, people over 70 years of age walk an average of 4740 steps per day, markedly fewer than other age groups [6]. As a countermeasure, the Health Japan 21 [2] and the 2013 publication of physical activity standards for health promotion [7] proposed inclusion of indicators that reflect not only the number of steps taken, but also the intensity and duration of activity measured as total expended energy during physical activities and their metabolic equivalents (METs). Previous research suggested associations between pedometer/accelerometer measurements of habitual physical activity and many aspects of physical and psychosocial health in older people [4]. These standards and measures suggested establishing physical activity guidelines for older people, to improve their physical activity levels.

One MET is defined as the amount of oxygen consumed while sitting at rest and is equal to 3.5 mL O^2^/kg body weight per minute. Inactivity among adults including older age groups accounts for 9.4% of all deaths worldwide [8]. It is also possible that inactivity may be a risk factor for weakening or decreased independence [9]. Muscle strength, motor function [10], depression [11], and respiratory function [12] have been reported as factors related to levels of physical activity in the older individuals. It has also been reported that when older people monitor their number of steps and acceleration, their levels of physical activity increase [13]. We believe that understanding more about muscle strength, motor function, diet, and activity levels in healthy people could help establish indicators for healthy life expectancy.

The people aged between 65–74 years are defined as “early elderly” and those 75 years and older as “late elderly” in Japan [14]. Many Japanese aged 65 years or older are in good health, just as they were during their middle-aged years. A long-lived country in Japan, we considered that factors to extend healthy life expectancy (muscular strength, motor function, diet, and activity level) could be compared by comparing early elderly and late elderly people. The proportion of the total older population has continued to increase since then, reaching 26.7% in 2016 (33.92 million) [5]. Currently, individuals who are 75 years and older comprise 12.9% of the total population [4]. Further self-management for optimization of health is required for older people. Therefore, we investigated healthy individuals of age 65 years and older who are spending their daily lives on health.

In Japan, a previous study showed that walking decreases during the winter among healthy older people living in snowy areas [15]. The physical activity of older individuals can be affected by the seasons. In the winter, snow accumulation makes walking difficult, resulting in increased risk of falling [13]. In environmental circumstances specific to the area, such as winter snowfall and road surface freezing, not only older adults, but also young people, can easily fall. Accordingly, it is often accidental, as one cannot avoid a fall due to body function only. There are few studies that have reported activity levels for healthy older people in the northern regions of Japan, such as Hokkaido, that experience a severe winter with heavy snow. We believe that knowing the types of activities needed to maintain muscle mass during the snow season is an indicator of self-management for optimizing health. In particular, focusing on elderly people whose activity is considered to be reduced, and knowing the characteristics of the activity levels of early elderly people and late elderly people will lead to obtaining an index for extending healthy life expectancy.

The aim of the study was to investigate age, daily activity level, and body composition of healthy older individuals when physical activity is limited during the snowy winter.

## 2. Materials and Methods

### 2.1. Subjects

Iwamizawa is a city in Hokkaido, the northernmost prefecture of Japan and a snowy, rural area in winter. The city of Iwamizawa has an estimated population of 83,383 people (38,973 men and 44,410 women) and a density of 175 persons per square kilometer [16]. This area received several heavy snowfalls leading to an interruption in the transportation of cars for a long period of time during the study period, resulting in significant obstacles in the lives of the subjects. Physical activity was limited due to the snow. The average life expectancy in Iwamizawa is 86.6 years for women and 79.0 years for men. The average healthy life expectancy is 73.18 and 69.88 years for women and men, respectively [16]. Iwamizawa is aging at a rate of 31.3%. Our subjects were 409 residents aged 20 years and older who participated in voluntary health checks, including a survey during the snowy period from January 2016 to March 2016. The survey was conducted at a drugstore used by residents for regular shopping. We targeted healthy residents who came to the drugstore and were interested in the study. A cross-sectional study involving 124 men and women aged 65 years and older who participated in a population survey was conducted.

### 2.2. Measurements and Survey Items

Activity levels were measured using an activity meter (Active style ProHJA-350IT; OMRON, Kyoto, Japan). We asked subjects to wear the activity meter on their waist for 14 days, as much as possible from the time they wake up until bedtime, except while bathing. After the 14-day period, we collected the activity meters and recorded the data. Data included the duration for which the meter was worn, number of steps, and METs when walking, measured as walking exercise recorded as METs × time. Height was measured using a digital height meter (AD-6400; AND, Tokyo, Japan). Body composition was measured using a body composition meter (Well Scan 900; Canon, Tokyo, Japan). Weight, total body fat, muscle mass, and combined fat and muscle mass per site were measured. Grip strength was measured using a force-measuring instrument (JAMAR hydraulic crush force meter; Fabrication Enterprises, Inc., White Plains, NY, USA).

A blood draw was performed using the self-drawn simple blood draw set (Eiken Chemical Co., Ltd., Tokyo, Japan). Blood specimens were collected and data recorded for total cholesterol, low density lipoprotein (LDL) cholesterol, high density lipoprotein (HDL) cholesterol, neutral fat, creatinine, urea nitrogen, uric acid, and hemoglobin A1c. Participants completed a self-administered Brief Diet History Questionnaire (BDHQ). This questionnaire has been developed for Japanese people. BDHQ assesses amount of nutrients habitually consumed from foods usually eaten (exclude intake from dietary supplements) as a unit of individual, in relatively easy way. BDHQ is designed to obtain each individual’s information of nutrient intake, food intake, as well as information of few markers of dietary behaviors. BDHQ is a 2-pages (A3 size) or 4-pages (A4 size), consisting of 102 questions. BDHQ has been developed to aim for uses in large-scale epidemiologic research and research with topics other than nutrition for a large number of people. The structure of the BDHQ and its validation was written in a published paper [17,18]. Using the BDHQ, we obtained data regarding the calorie intake and the intake of 3 major nutrients (carbohydrates, proteins, and fat) for 1 day.

### 2.3. Analysis Methods

Participants were divided into two groups according to age: those 65–74 years and those 75 and older. The number of steps was divided into ≥4000 steps group and <4000 steps group based on a standard of approximately 4000 steps, which is the average number of steps taken during 1 day for older Japanese women. Activity intensity was divided into ≥3METs and <3METs based on a standard of 3.0 METs, recommended by the World Health Organization as daily exercise intensity [19]. To test for significant differences, two independent groups of comparison tests were used. Results are shown as the average value ± standard deviation. We used the statistical software JMP 12.2.0 (SAS Institute Japan, Tokyo, Japan) and set significance level at less than 5%.

### 2.4. Ethical Considerations

This study was conducted with the approval of the Ethics Review Committee of the Graduate School of Health Sciences, Hokkaido University (approval number 15–96). We explained the research objectives and program, emphasizing that participation was voluntary, that there were no disadvantages to declining participation even midway through the study, and ensured that each individual understood to mutual satisfaction that the data obtained were used for research only and that all records were confidential and securely stored.

## 3. Results

Of the 124 participants aged ≥ 65 years who were analyzed, 84 were women and 40 were men. Average ages for subjects were 70.7 ± 4.75 for women and 72.4 ± 5.86 for men. Average numbers of steps were 4466.5 ± 2536.13 (women) and 5150.5 ± 2636.63 (men). Average METs when walking was 2.7 ± 1.67 METs (women) and 3.3 ± 1.64 METs (men). The body mass index (BMI) values were 22.9 ± 3.56 (women) and 23.5 ± 2.87 (men).

### 3.1. Comparison of Activity Level and Body Composition Between the 65–74 Age Group and the ≥75 Age Group

As shown in Table 1, the mean age of women in the 65–74 age group was 68.6 ± 2.97 years; for the ≥75 years age group, the mean was 77.7 ± 2.43 years. For the men in the 65–74 age group, the mean age was 68.5 ± 2.62 years; for the ≥75 age group, the mean was 78.9 ± 3.37 years. No difference was seen in activity levels between the women and the men.

Regarding body composition, there were differences in mean women’s heights between the age groups: for those 65–74, the average was 152.0 ± 4.92 cm; for those ≥ 75, 149.1 ± 4.95 cm (*p* = 0.03). BMI for those 65–74 years was 22.4 ± 3.48 kg/m^2^; for those ≥75 years: 24.4 ± 3.44 kg/m^2^ (*p* = 0.03). No difference was seen in physical activity, body composition, and blood test between the women’s and the men’s groups.

As shown in Table 1, in grip strength both men and women in the ≥75 age group showed lower values. Women in the 65–74 age group achieved 23.5 ± 4.38 kg; the ≥75 group’s results were 21.4 ± 3.24 kg; *p* = 0.03. The average value for men in the 65–74 age group was 39.2 ± 6.10 kg, and that for those in the ≥75 age group was 32.8 ± 5.33 kg (*p* < 0.01).

Nutritional conditions showed significant differences between the two age groups for 4 items for women and 1 item for men. For women, these were 1-day total energy intake in the 65–74 group, averaging 1586.2 ± 514.17 kcal and in the ≥75 group, the average intake was 1975.1 ± 563.09 kcal (*p* = 0.01); water intake for the 65–74 group was 1739.0 ± 486.07 g and that for those in the ≥75 group was 2025.2 ± 654.40 g (*p* = 0.04); protein intake in the 65–74 group was 69.8 ± 27.29 g and that for the ≥75 group was 96.7 ± 31.31 g (*p* < 0.01); and fat intake for the 65–74 group and the ≥75 group was 46.9 ± 19.19 g and 64.2 ± 24.88 g (*p* < 0.01), respectively. For men, there was a difference in water intake: those in the 65–74 age group and the ≥75 age group consumed 1819.9 ± 401.95 g and 2135.9 ± 536.45 g (*p* = 0.04), respectively.

Although blood analyses data for both men and women were within the standard ranges for Japanese people, women in the 65–74 age group had a higher value for total cholesterol at 222.5 ± 28.74 mg/dL than those in the ≥75 group, with an average value of 204.9 ± 34.34 mg/dL (*p* = 0.04). The average LDL cholesterol for 65–74 year old women was 125.9 ± 22.47 mg/dL and that for women ≥ 75 years was 110.3 ± 34.06 mg/dL (*p* = 0.03). For men, no significant differences were found between age groups for the remaining data.

### 3.2. Comparison of Activity Levels of Healthy Older Individuals

Because there was no difference in activity levels between the 65–74 and ≥75 age groups, we examined the other characteristics of older people. Further analyses were performed on pooled data. Participants were divided into two groups based on the standard of approximately 4000 steps, which is the average number of steps taken during 1 day for older Japanese person. These were then compared (Table 2). For women, a significant difference was seen in many of the body composition items. For men, there was no difference in any item. For women in the group who walked more than 4000 steps, BMI was within Japan’s standard range (18–24.9 kg/m^2^), although it was low in that range: 21.8 ± 2.79 kg/m^2^. For body fat and muscle mass by site for the group who walked more than 4000 steps, fat mass was small: women in the group who walked <4000 steps had fat mass of 16.6 ± 5.55 kg; women in the group who walked ≥4000 steps had fat mass of 13.0 ± 3.84 kg (*p* < 0.01). The left and right leg muscle mass measures were maintained (women, left leg, <4000 steps group had 6.9 ± 1.46 kg; for those walking ≥4000 steps, 6.6 ± 0.94 kg; for the right leg, less than 4000 steps group measured muscle mass of 6.6 ± 0.82 kg and those who walked more than 4000 steps group averaged 6.4 ± 0.75 kg of muscle mass).

For activity intensity level, we divided subjects into two groups, based on the standard of 3.0 METs recommended for daily exercise intensity by the World Health Organization. The results of the comparison are shown in Table 3. Regarding body composition among women, differences were found between the two groups for height (≥3.0 METs were of 147.5 ± 5.13 cm in height; <3.0 METs group measured 151.9 ± 4.83 cm in height (*p* = 0.01). For muscle mass, the ≥3.0 METs group had 31.8 ± 2.14 kg overall, while the <3.0 METs group had 34.9 ± 4.12 kg (*p* = 0.02). For men, there was no significant difference in height and muscle mass; the ≥3.0 METs group showed lower values for body fat (body fat percentage: ≥3.0 METs group had 17.5 ± 6.79% and the <3.0 METs group had 22 ± 4.90% (*p* = 0.03). Analysis of body fat mass by site revealed that the ≥3.0 METs group had lower values at all sites, the left and right arms (left arm: *p* = 0.03; right arm: *p* = 0.02); torso (*p* = 0.03); left and right legs: left leg: *p* = 0.04; right leg: *p* = 0.04.

## 4. Discussion

The average age of our participants was younger than Japan’s average life expectancy and closer to that of the average healthy life expectancy (women: 70.7 ± 4.75 years; men: 72.4 ± 5.86 years).

### 4.1. Comparison of Activity Levels and Body Composition Between the 65–74 Age Group and the ≥75 Age Group

Age and muscle mass have been reported as factors that influence the number of steps and expended energy during physical activity [20]. According to Roberts and Dallal [21], for both men and women, with ageing, the total number of expended calories reduced with a decrease in muscle mass and physical activity levels. The results of our study indicated that muscle mass and activity levels of the ≥75 age group were maintained at the same levels as for those 65–74 years of age. No differences between the groups were found regarding total expended calories. Therefore, there was no actual difference according to age regarding activity levels during snowy periods for healthy older people living in these areas.

However, in this study, a difference according to age was found regarding grip force. It has been reported that there is a decrease in grip force with ageing, reflecting change in muscle strength [22], corroborating our results.

### 4.2. Activity Level (Number of Steps and Activity Intensity)

The nationwide average numbers of steps in Japan for men and women aged ≥65 years are 5779 and 4736, respectively [4]. In this survey, the number of steps recorded was less than the national average. With advancing age, the amount of daily activity is less and the elderly spend more time indoors during the day [23]. Falling snow and high winds lead to reduced visibility and loss of balance. Inconsistent changes in the condition of the road and sidewalk surfaces where the snow and ice melts at daytime and refreezes at night increases the chances of slipping [24]. The city of Iwamizawa, on the northernmost island of Japan, is located between longitudes 141°47′E and 43°12′N. The annual mean temperature ranges between 6 °C and 10 °C (the average temperature in August is approximately 22 °C, while the average temperature in January is −12 °C to –4 °C) [25]. The island is covered with snow or ice for one-third of the year, due to heavy snowfall during the winter. Many older people experience falls every year, and walking is especially difficult when there is snowfall in Hokkaido [23]. According to Aoyagi and Shephard [26], the number of steps taken by older people peaks when the average temperature is approximately 17 °C and decreases as the weather becomes colder. Our results showed that it is possible that walking is difficult during winter (the outside temperature during our study period was approximately 1 °C). However, because the SD range is wide, a high level of individual experience is apparent. In addition, the results indicating that people walking ≥4000 steps expended more calories support the data indicating a correlation between number of steps and amount of energy expended during physical activity. In other words, the number of steps explains the amount of energy expended by healthy older individuals.

Men generally performed more physical activity than women [27]. In that study, this was primarily due to ageing-related phenomena of medium-intensity activities; in contrast, among women, decreases were for low-intensity activities. Our results indicated that there was no difference between groups when we compared the number of steps for men; however, there was a difference between groups when we compared the activity intensity of 3.0 METs. For women, there was a difference between groups when we compared the number of steps, in spite of no difference when we compared the activity intensity of 3.0 METs. Therefore, in our survey, there was a gender difference, according to the evaluation of activity level items, as reported in previous studies.

According to the Physical Activity Standards for Health Promotion 2013 [5], the suggested exercise standards for people aged 65 years and older include performance of physical activity at an intensity of daily exercise, regardless of METs. The results of this study targeting healthy older people clarified that even during the winter in snowy areas, walking more than 4000 steps with an exercise intensity of more than 3.0 METs may be beneficial for healthy aging. This is the first study to investigate the physical activity of healthy older adults living in a heavy snowfall area in Japan. The results suggest that physical activity may promote health even in the winter, when it is difficult to walk and exercise due to the snow. These results may be used to inform guidelines regarding extending healthy life expectancy by means of physical exercise.

### 4.3. Limitations

The limitations of this study include its relatively small sample size, and a lack of detailed data on certain measures of interest, specifically measures of nutrition and nutritional deficiencies. The small sample size may have been due to the study being conducted during the winter, when walking outside was difficult due to heavy snowfalls. We believe that the analysis that built the regression model is important. However, the variance inflation factor (VIF) was 10 or more. Therefore, the regression model was not considered in consideration of multicollinearity between variables. Future studies should recruit larger samples of older adults and should include more variables, including more detailed data on measures of nutritional status and nutritional deficiencies. Randomized controlled interventions are required for further investigation of the association between of physical activity and body composition in older Japanese adults.

## 5. Conclusions

The following observations became clear after investigating the activity levels during snowy periods and body composition of older people living in snowy rural areas. First, after comparing the 65–74 age group and the ≥75 age group, no significant difference was found in activity levels among men and women. For body composition among women, there were differences in height and BMI. For men, no difference was found between the two age groups. Second, healthy older women living in snowy areas who walk more than 4000 steps/day had less fat mass and maintained more muscle mass compared to those walking less than 4000 steps each day. Finally, healthy older men living in snowy areas who had activity intensity levels of more than 3.0 METs had lower body fat percentages compared to those in the under 3.0 METs group.

## Figures and Tables

**Table 1 ijerph-17-05316-t001:** Comparison of activity level and body composition between the 65–74 age group and the ≥75 age group.

Activity Level and Body Composition		Women					Men				
		65–74 (*n* = 19)		≥75 (*n* = 65)			65–74 (*n* = 15)		≥75 (*n* = 25)		
		Mean	SD	Mean	SD	*p*	Mean	SD	Mean	SD	*p*
**Age**	Age (years)	68.6	2.97	77.7	2.43	<0.01	68.5	2.62	78.9	3.37	<0.01
Activity level	Activity Meter Installation Period (Day)	12.8	2.01	12.7	1.73	n.s.	13.1	1.93	12.3	3.11	n.s
	Number of Steps (Steps)	4462.2	2006.48	4481.2	3917.1	n.s.	5473.8	2463.05	4611.7	2910.03	n.s
	Walk Distance (m)	3	1.37	3.1	2.8	n.s.	3.9	1.76	3.2	2	n.s
	Calories When Walking (kcal)	94.1	50.38	99	98.21	n.s.	160.1	76.82	126.7	96.25	n.s
	METs When Walking (METs)	1.5	1.24	1.6	1.91	n.s.	1.9	1.34	1.9	1.96	n.s
	Total Expended Calories (kcal)	1576.4	220.25	1590.7	252.22	n.s.	1973	237.07	1887.8	142.5	n.s
Body composition	Height (cm)	152	4.92	149.1	4.95	0.03	164.2	6.12	161.5	5.54	n.s
	Weight (kg)	51.6	7.46	54.3	7.62	n.s.	63.5	9.07	60.8	5.4	n.s
	Body Fat Percentage (%)	26.9	5.99	30	6.3	n.s.	21.2	5.97	19.7	5.59	n.s
	Body Fat Mass (kg)	14.2	5.15	16.3	4.45	n.s.	13.8	4.85	12.2	4.23	n.s
	Muscle mass (kg)	34.4	3.17	34.9	6.32	n.s.	46	5.16	45	3.11	n.s
	B.M.I. (kg/m^2^)	22.4	3.48	24.4	3.44	0.03	23.5	2.8	23.4	3.15	n.s
Grip strength	Grip strength (kg)	23.5	4.38	21.4	3.24	0.03	39.2	6.1	32.8	5.33	<0.01
Dietary intake of participants derived from the BDHQ	Energy (kcal)	1586.2	514.17	1975.1	563.09	0.01	1799.8	497.18	2015.5	530.14	n.s
	Water (g)	1739	486.07	2025.2	654.4	0.04	1819.9	401.95	2135.9	536.45	0.04
	Protein (g)	69.8	27.29	96.7	31.31	<0.01	74	31.52	84.2	26.36	n.s
	Fat (g)	46.9	19.19	64.2	24.88	<0.01	48.2	17.2	49.3	15.27	n.s
	Carbohydrates (g)	211.8	74.16	242.3	82.42	n.s.	241.1	70.88	265.2	81.87	n.s
	Alcohol (g)	3.7	10.75	3.7	10.01	n.s.	12.2	20.62	21.7	44.11	n.s
Blood test	Total cholesterol (mg/dL)	222.5	28.74	204.9	34.34	0.04	188.3	37.17	179.7	26.95	n.s
	LDL cholesterol (mg/dL)	125.9	22.47	110.3	34.06	0.03	103.4	37.96	96.3	21.37	n.s
	HDL cholesterol (mg/dL)	67	16.51	63.4	13.32	n.s.	52.3	19.24	55.1	16.42	n.s
	Triglycerides (mg/dL)	138.5	91.33	117.3	70.12	n.s.	147.6	78.34	133.4	83.15	n.s
	HbA1c (%)	5.8	0.5	5.8	0.43	n.s.	5.9	0.66	6	0.53	n.s

BDHQ: Brief-Type Self-Administered Diet History Questionnaire. n.s.: not significant. *p*-Values were derived using t-test.

**Table 2 ijerph-17-05316-t002:** Comparison by number of steps.

Activity Level and Body Composition		Women					Men				
		Less than 4000 Steps (*n* = 36)		Over 4000 Steps (*n* = 48)			Less than 4000 Steps (*n* = 18)		Over 4000 Steps (*n* = 22)		
		Mean	SD	Mean	SD	*p*	Mean	SD	Mean	SD	*p*
**Age**	Age (years)	70.6	4.74	70.7	4.82	n.s.	73.6	6.66	71.3	4.94	n.s.
Activity Level	Activity Levels During Meter-Wearing Period (days)	12.4	1.87	13.1	1.96	n.s.	12.2	3.25	13.4	1.12	n.s.
	Number of Steps (Steps)	2576.7	1027.22	6268.5	2212.46	<0.01	2812.7	675.82	7265.6	1804.33	<0.01
	Walking Distance (m)	1.7	0.71	4.2	1.57	<0.01	2	0.51	5.2	1.21	<0.01
	Calories When Walking (kcal)	49.6	24.57	138.8	58.84	<0.01	79	25.63	209.7	70.51	<0.01
	Mets When Walking (METs)	0.6	0.46	2.4	1.44	<0.01	0.6	0.36	3.1	1.3	<0.01
	Total Expended Calories (kcal)	1527	176.11	1629.8	257.74	0.04	1901.5	211.97	1976.8	204.43	n.s.
Body Composition	Height (cm)	151.8	5.07	151	5.05	n.s.	162.6	4.89	163.8	6.82	n.s.
	Weight (kg)	55.1	8.19	49.6	5.76	<0.01	64.7	7.52	60.9	8.31	n.s.
	Body Fat Percentage (%)	29.6	6.33	25.8	5.48	<0.01	22.1	5.14	19.6	6.19	n.s.
	Body Fat Mass (kg)	16.6	5.55	13	3.84	<0.01	14.6	4.58	12.2	4.56	n.s.
	Muscle mass (kg)	35.4	4.59	33.8	3.37	n.s.	46.4	3.92	45.1	5.05	n.s.
	B.M.I. (kg/m^2^)	24	3.95	21.8	2.79	<0.01	24.5	2.8	22.7	2.74	n.s.
(Body Fat by Site)	Body Fat Left Arm (kg)	1	0.34	0.8	0.26	<0.01	0.9	0.26	0.8	0.3	n.s.
	Body Fat Right Arm (kg)	1.1	0.39	0.9	0.31	0.01	0.9	0.25	0.8	0.31	n.s.
	Body Fat Torso (kg)	8.5	2.88	6.7	1.98	<0.01	7.5	2.37	6.3	2.33	n.s.
	Body Fat Left Leg (kg)	2.9	1.07	2.3	0.7	<0.01	2.6	0.86	2.1	0.83	n.s.
	Body Fat Right Leg (kg)	3	1	2.3	0.7	<0.01	2.7	0.85	2.2	0.8	n.s.
Muscle Mass by Site	Muscle Mass Left Arm (kg)	2.4	0.44	2.2	0.29	0.03	3.1	0.28	3.1	0.37	n.s.
	Muscle Mass Right Arm (kg)	2.3	0.49	2.1	0.31	0.05	3.0	0.32	3	0.38	n.s.
	Muscle Mass Torso (kg)	17.2	1.9	16.5	1.53	0.05	23.0	1.82	22.3	2.37	n.s.
	Muscle Mass Left Leg (kg)	6.9	1.46	6.6	0.94	n.s.	8.2	2.25	8.4	1.05	n.s.
	Muscle Mass Right Leg (kg)	6.6	0.82	6.4	0.75	n.s.	8.4	0.73	8.1	1.47	n.s.
Grip Strength	Grip strength (kg)	23.4	3.51	22.8	4.83	n.s.	36.3	7.79	37.4	5.42	n.s.
Dietary Intake of Participants Derived from the BDHQ	Energy (kcal)	1745.5	573.14	1606.1	518.76	n.s.	1809.7	477.74	1944.9	548.3	n.s.
	Water (g)	1871.2	501.15	1739.5	569.54	n.s.	1886.8	540.03	1985.1	418.15	n.s.
	Protein (g)	81.2	30.57	70.8	29.41	n.s.	69.7	22.17	85.2	34.11	n.s.
	Fat (g)	55.9	24.27	46	17.91	0.04	46.1	15.62	50.9	16.97	n.s.
	Carbohydrates (g)	221	80.68	216.5	73.53	n.s.	251.3	72.45	249	79.15	n.s.
	Alcohol (g)	2.9	9.07	4.5	11.8	n.s.	12.3	21.46	18.8	38.47	n.s.

BDHQ: Brief-Type Self-Administered Diet History Questionnaire. n.s.: not significant. *p*-Values were derived using t-test.

**Table 3 ijerph-17-05316-t003:** Comparison by activity intensity.

Activity Level and Body Composition		Women					Men				
		Over 3 METs (*n* = 11)		Less than 3 METs (*n* = 73)			Over 3 METs (*n* = 11)		Less than 3 METs (*n* = 29)		
		Mean	SD	Mean	SD	*p*	Mean	SD	Mean	SD	*p*
**Age**	Age (years)	73.7	5.12	70.3	4.58	0.31	72.6	5.33	72.3	6.14	n.s.
Activity Level	Activity Meter Installation Period (Day)	13.4	1.07	12.7	2.02	n.s.	13.5	0.52	12.6	2.81	n.s.
	Step Count (Steps)	8882.6	2898.66	3869.7	1801.32	<0.01	8156.5	1817.09	4010.3	1899.34	<0.01
	Walk Distance (m)	6	2.17	2.6	1.25	<0.01	5.7	1.26	2.9	1.39	<0.01
	Calories When Walking (kcal)	197.1	69.6	81.5	48.98	<0.01	230.4	80.64	116.2	63.44	<0.01
	Mets When Walking (METs)	4	1.72	1.2	0.97	<0.01	3.3	1.67	1.4	1.17	<0.01
	Total Expended Calories (kcal)	1622.4	189.92	1573.9	231.35	n.s.	1955.5	188.11	1935.6	219.11	n.s.
Body Composition	Height (cm)	147.5	5.13	151.9	4.83	0.01	163.7	4.28	163.2	6.65	n.s.
	Weight (kg)	48.3	4.62	52.8	7.72	n.s.	58.6	8.08	64.3	7.64	n.s.
	Body Fat Percentage (%)	28.3	5.97	27.5	6.23	n.s.	17.5	6.79	22.1	4.9	0.04
	Body Fat Mass (kg)	13.8	3.81	14.8	5.21	n.s.	10.6	4.66	14.4	4.26	0.03
	Muscle Mass (kg)	31.8	2.14	34.9	4.12	0.02	44.6	4.65	46.1	4.55	n.s.
	B.M.I. (kg/m^2^)	22.3	2.77	22.9	3.66	n.s.	21.9	2.74	24.2	2.7	0.03
(Body Fat by Site)	Body Fat Left Arm (kg)	0.9	0.24	0.9	0.33	n.s.	0.7	0.3	0.9	0.26	0.03
	Body Fat Right Arm (kg)	0.9	0.24	1	0.38	n.s.	0.7	0.31	0.9	0.26	0.02
	Body Fat Torso (kg)	7.1	1.97	7.6	2.7	n.s.	5.4	2.38	7.4	2.2	0.03
	Body Fat Left Leg (kg)	2.4	0.71	2.6	0.98	n.s.	1.9	0.83	2.5	0.81	0.04
	Body Fat Right Leg (kg)	2.5	0.69	2.6	0.95	n.s.	1.9	0.85	2.6	0.79	0.04
(Muscle Mass by Site)	Muscle Mass Left Arm (kg)	2.1	0.17	2.3	0.39	0.05	3	0.34	3.1	0.33	n.s.
	Muscle Mass Right Arm (kg)	2	0.18	2.2	0.43	n.s.	3	0.35	3	0.36	n.s.
	Muscle Mass Torso (kg)	15.6	1.05	17	1.75	0.01	22	2.12	22.8	2.14	n.s.
	Muscle Mass Left Leg (kg)	6.2	0.55	6.8	1.27	n.s.	8.4	1.05	8.3	1.89	n.s.
	Muscle Mass Right Leg (kg)	6	0.47	6.5	0.8	0.02	7.7	1.73	8.4	0.85	n.s.
Grip Strength	Grip Strength (kg)	21.6	2.95	23.3	4.36	n.s.	34.9	4.55	37.8	7.13	n.s.
Dietary Intake of Participants Derived from the BDHQ	Energy (kcal)	1576.2	471.17	1687.4	558.17	n.s.	1989.3	657.93	1839.5	454.97	n.s.
	Water (g)	1562.3	483.85	1836.4	539.73	n.s.	1962.6	528.99	1929.2	464.28	n.s.
	Protein (g)	69.8	32.23	76.7	30.12	n.s.	78.9	26.64	77.4	31.29	n.s.
	Fat (g)	42.5	16.29	51.9	22.18	n.s.	47.8	16.35	48.9	16.58	n.s.
	Carbohydrates (g)	224.6	59.26	217.9	79.02	n.s.	267.6	95.32	243.5	66.65	n.s.
	Alcohol (g)	0.7	1.84	4.1	11.14	n.s.	21.5	51.68	13.6	19.91	n.s.

BDHQ: Brief-Type Self-Administered Diet History Questionnaire. n.s.: not significant. *p*-Values were derived using t-test.

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
