# Peer review of "Association between Physical Activity Levels and Body Composition among Healthy Older Japanese Adults during a Snowy Winter: A Cross-Sectional Study"

_ijerph, 2020, doi:10.3390/ijerph17155316_

Round 1

Reviewer 1 Report

The work is presented in a professional and clear way. The topic is very interesting and extremely important nowadays: the results are certainly useful for formulating guidelines and public policies (lines 261-263).

I would like to make a few comments, hoping to enrich the work and contribute to the clarity of the article:

  • the context is well described, however, how does the snow context affect the uniqueness of the research and the results obtained? It is only in the fact that the " the number of steps recorded was less than the national average" (line 233)? The explanation in “4.2. Activity Level (Number of Steps and Activity Intensity)” deserves to be better explained by highlighting the uniqueness of the context. How does the context impact on the healthy life expectancy? In particular, how can you relate the context’s characteristics to the result described in the abstract "For healthy older Japanese individuals in this snowy winter region, walking> 4000 steps daily for women and exercise of> 3.0 METs for men may indicated health promoting activities " (line 24-25)? Provocatively: if this same work had been done in Tokyo, what variables would have been different for the difference of the context? Could you highlight these points?
  • Saying “To understand determinants of healthy life expectancies in older Japanese individuals in a snowy winter region” (line 17) could be better said “to contribute in understanding some determinants …” (or something similar) because the research isn’t understanding all the aspects that determine the healthy life expectancies.
  • There is a discrepancy defining the reason why 4000 steps: is the average among women (line 124) or among older Japanese persons (line 185)?

I hope these suggestions could be useful.

Best regards

Author Response

July 14, 2020

Prof. Dr. Paul B. Tchounwou

Editor-in-Chief

Dear Editor:

Thank you very much for reviewing our manuscript " Association between Physical Activity Levels and Body Composition among Healthy Older Japanese Adults during a Snowy Winter: Across-Sectional Study" (IJERPH-862252).

We have listed your comments in the review of IJERPH-862252 again and explained how the manuscript was improved, item for item. We hope the following answers are comprehensive for you to consider publication of this manuscript in the International Journal of Environmental Research and Public Health.

Response 1

Point 1: I found that sections 2 & 3 should be re‐organized and be shortened. It may be easier for the readers if the authors define properly the mixture of regression model and the class‐ membership equation first before moving to the computation of the GINI and of the Polarization of subgroups. Sections 2.1 and 2.2 are too long and can be significantly reduced. In section 2.1 the authors assume the condition uk > uj, but this does not appear anywhere else in the calculation of the mixture of regression model. After equation (10) all the other equations are not numbered.

Answer to comment (point 1)

We have re-organized and shortened sections 2 and 3.

I believe that the analysis that built the regression model is important. However, the variance inflation factor (VIF) of the measurement items was 10 over. Therefore, the regression model was not considered in consideration of multicollinearity between variables. In the future, we would like to consider after accumulating data. This content was added to the limitations of research (281line).

Response 2

Point 2: The probability for a given country h to be in a class k should be the proportion of observations (households) in country h that belong to the income class k. On page 9, the first equation (it would be easier for the reader if the equation is numbered) is not exactly the proportion of people because the authors take the sum of the probability. The interpretation of the equation in not obvious. Normally, after estimating a mixture of regression model we have for each observation its estimated probabilities to be classified into the different classes identified. What is often done is to classify a given observation into the class where its estimated probability is higher. In many software this is also the method used that gives us the proportion of people in each of the classes. The authors should explain the equation on page 9 and how to interpret it. Alternatively, they may use the proportion approach which will make the interpretation easier.

Answer to comment (point 2)

Thank you for your suggestion that they may use the proportion approach which will make the interpretation easier. Certainly, we think that a clearer consideration is possible by the test of ratio difference. This report presented data on special heavy snowfall areas that have never been published in Japan. Although there are few reports that can be compared with this region, I reconsidered the reference for consideration and added a paper considering the test of ratio difference to the consideration (Reference number 23).

Response to Reviewer 1 Comments

  • the context is well described, however, how does the snow context affect the uniqueness of the research and the results obtained? It is only in the fact that the "the number of steps recorded was less than the national average" (line 233)? The explanation in “4.2. Activity Level (Number of Steps and Activity Intensity)” deserves to be better explained by highlighting the uniqueness of the context. How does the context impact on the healthy life expectancy? In particular, how can you relate the context’s characteristics to the result described in the abstract "For healthy older Japanese individuals in this snowy winter region, walking> 4000 steps daily for women and exercise of> 3.0 METs for men may indicated health promoting activities " (line 24-25)? Provocatively: if this same work had been done in Tokyo, what variables would have been different for the difference of the context? Could you highlight these points?

Answer to comment 1

I described the uniqueness of the study on the context of snow and its effect on the results obtained (line243). We thought that maintaining a healthy life expectancy in a heavy snowfall region was maintaining the amount of activity even if snowfall made walking difficult. We added to the discussion about the snow context affect the uniqueness of the research. The present study had several limitations. I believe that the analysis that built the regression model is important. However, the variance inflation factor (VIF) of the measurement items shown in Table 1 was 10 over. Therefore, the regression model was not considered in consideration of multicollinearity between variables. In the future, we would like to consider after accumulating data. This content was added to the limitations of research (line281).

  • Saying “To understand determinants of healthy life expectancies in older Japanese individuals in a snowy winter region” (line 17) could be better said “to contribute in understanding some determinants …” (or something similar) because the research isn’t understanding all the aspects that determine the healthy life expectancies.

Answer to comment 2

As mentioned above, we edited the text in the methods (line17).

  • There is a discrepancy defining the reason why 4000 steps: is the average among women (line 124) or among older Japanese persons (line 185)?

Answer to comment 3

As suggested, we have edited the reason for 4000 steps in the discussion (line185).

We hope that this manuscript can be reviewed again and considered for publication in “the International Journal of Environmental Research and Public Health”.

Very sincerely yours,

Katsuhiko Ogasawara

Faculty of Health Sciences, Hokkaido University

Kita12, Nishi5, Kita-ku, Sapporo, Hokkaido 060-0812, Japan

Phone/Fax: +81(11)706-3409

Email: oga@hs.hokudai.ac.jp

Reviewer 2 Report

Thank you for providing the opportunity to review your work.

This paper investigates age, daily activity level, and body composition of healthy older individuals when physical activity is limited during the snowy winter.

Please see my comments/suggestions.

Regarding 1. Introduction: where is research gaps? For example, there is no research regarding physical activity of older adults’ in cold area (long winter with lot of snow)?

Lines 30-31 and 57-60: The part (lines 57-60) should be incorporated with the first part of introduction (lines 30-31). The readers may not have enough information about aging status of Japan.

Line 42: Why metabolic equivalents (METs) is important for healthy living? Without this statement, reader may assume the significance of METs.

Lines 90-91: need more detail about participants (age, gender, socio-economic status)

Line 114: It would be better to provide this questionnaire as a supplement.

Line 131: xxx is approval number?

Regarding 3. Results, it seems all measurements use mean to find differences. It might be necessary to provide reason for this method. Why median is not used? or other statistical methods such as comparing two groups using independence T-test or other methods?
Line 140:  Is average or mean should be single number? Does it show 95% CI?

Regarding 4. Discussion, it should stress the significance of the results of the work, not repeat them. Also, consider the followings in this chapter: any challenges? contribution of this study? what are the difference or similarity of other studies?

Regarding 5. Conclusions, this chapter should summary the combination between succinct results and essence of discussion, not repeating results.

Author Response

July 14, 2020

Prof. Dr. Paul B. Tchounwou

Editor-in-Chief

Dear Editor:

 Thank you very much for reviewing our manuscript " Association between Physical Activity Levels and Body Composition among Healthy Older Japanese Adults during a Snowy Winter: Across-Sectional Study" (IJERPH-862252).

 We have listed your comments in the review of IJERPH-862252 again and explained how the manuscript was improved, item for item. We hope the following answers are comprehensive for you to consider publication of this manuscript in the International Journal of Environmental Research and Public Health.

Response 1

Point 1: I found that sections 2 & 3 should be re‐organized and be shortened. It may be easier for the readers if the authors define properly the mixture of regression model and the class‐ membership equation first before moving to the computation of the GINI and of the Polarization of subgroups. Sections 2.1 and 2.2 are too long and can be significantly reduced. In section 2.1 the authors assume the condition uk > uj, but this does not appear anywhere else in the calculation of the mixture of regression model. After equation (10) all the other equations are not numbered.

Answer to comment (point 1)

We have re-organized and shortened sections 2 and 3.

I believe that the analysis that built the regression model is important. However, the variance inflation factor (VIF) of the measurement items was 10 over. Therefore, the regression model was not considered in consideration of multicollinearity between variables. In the future, we would like to consider after accumulating data. This content was added to the limitations of research (line 281).

Response 2

Point 2: The probability for a given country h to be in a class k should be the proportion of observations (households) in country h that belong to the income class k. On page 9, the first equation (it would be easier for the reader if the equation is numbered) is not exactly the proportion of people because the authors take the sum of the probability. The interpretation of the equation in not obvious. Normally, after estimating a mixture of regression model we have for each observation its estimated probabilities to be classified into the different classes identified. What is often done is to classify a given observation into the class where its estimated probability is higher. In many software this is also the method used that gives us the proportion of people in each of the classes. The authors should explain the equation on page 9 and how to interpret it. Alternatively, they may use the proportion approach which will make the interpretation easier.

Answer to comment (point 2)

Thank you for your suggestion that they may use the proportion approach which will make the interpretation easier. Certainly, we think that a clearer consideration is possible by the test of ratio difference. This report presented data on special heavy snowfall areas that have never been published in Japan. Although there are few reports that can be compared with this region, I reconsidered the reference for consideration and added a paper considering the test of ratio difference to the consideration (Reference number 23).

Response to Reviewer 2 Comments

Regarding 1.

Introduction: where is research gaps? For example, there is no research regarding physical activity of older adults’ in cold area (long winter with lot of snow)?

Lines 30-31 and 57-60: The part (lines 57-60) should be incorporated with the first part of introduction (lines 30-31). The readers may not have enough information about aging status of Japan.

Line 42: Why metabolic equivalents (METs) is important for healthy living? Without this statement, reader may assume the significance of METs.

Lines 90-91: need more detail about participants (age, gender, socio-economic status)

Line 114: It would be better to provide this questionnaire as a supplement.

Line 131: xxx is approval number?

Answer to comment 1

Introduction: This report presented data on special heavy snowfall areas that have never been published in Japan. Although there are few reports that can be compared with this region, I reconsidered the reference for consideration and added a paper considering the test of ratio difference to the consideration (line243).

Lines 30-31 and 57-60: We incorporated with the first part of introduction (lines 37-40). Thank you for pointing out that. We are very sorry for the inconsistency and deleted the comment from the part (line63).

Line 42: Thank you for your suggestions. I explained the reason that metabolic equivalents (METs) is important for healthy living (line 45).

Lines 90-91: Thank you for pointing out the fundamental problem. We have included detailed information about the participants in the results (line145-149).

Line 114: Thank you for your suggestions. It was difficult to provide the questionnaire as a supplement, because we were instructed to shorten the paper.

Line 131: Yes, it is. It means approval number. We added it (line137).

Regarding 3. Results, it seems all measurements use mean to find differences. It might be necessary to provide reason for this method. Why median is not used? or other statistical methods such as comparing two groups using independence T-test or other methods?

Line 140:  Is average or mean should be single number? Does it show 95% CI?

Answer to comment 3

The statistical analysis was comparing two groups using independence t-tests. We added the information in the table. Thank you for pointing out that. We are very sorry for the inconsistency and believe that the analysis that built the regression model is important. However, the variance inflation factor (VIF) of the measurement items shown in Table 1 was 10 over. Therefore, the regression model was not considered in consideration of multicollinearity between variables. This content was added to the limitations of research.

Regarding 4. Discussion, it should stress the significance of the results of the work, not repeat them. Also, consider the followings in this chapter: any challenges? contribution of this study? what are the difference or similarity of other studies?

Answer to comment 4

Thank you for pointing out that. We are very sorry for the inconsistency and re-edited as much as possible.

Regarding 5. Conclusions, this chapter should summary the combination between succinct results and essence of discussion, not repeating results.

Answer to comment 5

We are very sorry for the inconsistency and re-edited as much as possible. We reconsidered the limits of analysis of results and references (line 281).

We hope that this manuscript can be reviewed again and considered for publication in “the International Journal of Environmental Research and Public Health”.

Very sincerely yours,

Katsuhiko Ogasawara

Faculty of Health Sciences, Hokkaido University

Kita12, Nishi5, Kita-ku, Sapporo, Hokkaido 060-0812, Japan

Phone/Fax: +81(11)706-3409

Email: oga@hs.hokudai.ac.jp

Round 2

Reviewer 2 Report

Thank you for revising your manuscript with my previous comments.

The paper was improved and read better now.

One comment would be:

The way to reporting numbers and statistical results might need some work.

Also, still it would be better to include questionnaire as a independent document (supplement of this paper). It doesn't affect the length of manuscript.

Author Response

July 20, 2020

Prof. Dr. Paul B. Tchounwou

Editor-in-Chief

Dear Editor:

Thank you very much for reviewing our manuscript " Association between Physical Activity Levels and Body Composition among Healthy Older Japanese Adults during a Snowy Winter: Across-Sectional Study" (IJERPH-862252).

We have listed your comments in the review of IJERPH-862252 again and explained how the manuscript was improved, item for item. We hope the following answers are comprehensive for you to consider publication of this manuscript in the International Journal of Environmental Research and Public Health.

Response to Reviewer 2 Comments

Regarding 1.

Thank you for revising your manuscript with my previous comments.

The paper was improved and read better now.

One comment would be:

The way to reporting numbers and statistical results might need some work.

Also, still it would be better to include questionnaire as a independent document (supplement of this paper). It doesn't affect the length of manuscript.

Answer to comment 1

Thank you for your suggestions. We apologize for the lack of explanation in the last comment.

We contacted the developers of BDHQ (Brief Diet History Questionnaire). We received the answer that they are difficult to describe because BDHQ is protected by a patent. Therefore, we cannot submit as supplement, I added a paper showing evidence of validity to the method and explained it carefully (line112-119).

We hope that this manuscript can be reviewed again and considered for publication in “the International Journal of Environmental Research and Public Health”.

Very sincerely yours,

Katsuhiko Ogasawara

Faculty of Health Sciences, Hokkaido University

Kita12, Nishi5, Kita-ku, Sapporo, Hokkaido 060-0812, Japan

Phone/Fax: +81(11)706-3409

Email: oga@hs.hokudai.ac.jp